# Structures in G proteins important for subtype selective receptor binding and subsequent activation

Volker Jelinek [1], Nadja Mösslein [1] & Moritz Bünemann [1✉]

G protein-coupled receptors (GPCRs) selectively couple to specific heterotrimeric G proteins comprised of four subfamilies in order to induce appropriate physiological responses. However, structural determinants in Gα subunits responsible for selective recognition by approximately 800 human GPCRs have remained elusive. Here, we directly compare the influence of subtype-specific Gα structures on the stability of GPCR-G protein complexes and the activation by two Gq-coupled receptors. We used FRET-assays designed to distinguish multiple Go and Gq-based Gα chimeras in their ability to be selectively bound and activated by muscarinic M$_3$ and histaminic H$_1$ receptors. We identify the N-terminus including the αN/β1-hinge, the β2/β3-loop and the α5 helix of Gα to be key selectivity determinants which differ in their impact on selective binding to GPCRs and subsequent activation depending on the specific receptor. Altogether, these findings provide new insights into the molecular basis of G protein-coupling selectivity even beyond the Gα C-terminus.

[1] Institute of Pharmacology and Clinical Pharmacy, Philipps-University Marburg, Marburg, Germany. ✉email: moritz.buenemann@staff.uni-marburg.de

Gprotein-coupled receptors (GPCRs) are extensively studied drug targets that are addressed by about one-third of all approved drugs[1]. They serve as the largest human receptor family which comprises nearly 800 individuals. Remarkably, these couple to only 16 structurally similar Gα subunits that are assigned to four different G protein classes (Gs, Gi/o, Gq/11, G12/13) leading each to distinct cellular responses[2,3]. Typically, GPCRs preferentially interact with one specific G protein class, but promiscuity in G protein-coupling has also been observed[4,5]. Diverse expression of various Gα subtypes in a cell requires GPCRs to distinguish between them in order to ensure signaling via the proper pathway. This raises the question of which protein–protein recognition sites exist to enable coupling specificity. However, for some decades, no general mechanism has been found, presumably due to the fact that GPCR subfamilies independently evolved in parallel mechanisms to activate the same G protein during their evolutionary history[6]. To elucidate the molecular basis that underlies this G protein selectivity, structural studies, particularly cryo-EM structures of GPCR-G protein complexes, were successfully performed in recent years[7–20]. These structures were obtained with G proteins of the Gs, Gi/o, and Gq/11 family and provided overwhelming evidence that the C-terminal helix of the Gα subunit binds into the intracellular cavity of GPCRs and is thereby intimately involved in the mechanism responsible for G protein activation. Furthermore, they implied that the sizes of bulky Gs or small Gi C-termini affect the accommodation by distinct capacities of receptor binding pockets[19]. Apart from that, other interaction sites such as the αN/β1 and β2/β3 loop were identified to interact with regions near to the intracellular loops (ICLs) of the receptor, especially ICL2[21,22]. However, structures failed to reveal distinct interaction sites between GPCRs and different Gα subfamilies that were hypothesized as a Gα selectivity barcode[6]. In addition, structures generally provide only snapshots of nucleotide-free complexes which miss the entire temporal sequence of coupling events. For instance, initial intermediate states might already function as a selectivity filter[20,23]. Functional studies using classical second messenger readouts do not allow for a quantitative comparison of distinct signaling pathways necessary to verify structural elements in the G protein that are selectively recognized by GPCRs[24–28]. In addition, it is very hard to quantify the binding of G proteins to GPCRs in biochemical assays, in particular, without assessing kinetics. Even though a recent study revealed that Gi-coupled GPCRs are more specifically demanding for cognate Gi/o C-termini than other receptors, many GPCRs exert coupling specificity predominantly by structures in the Gα subunit which still remain elusive[29]. For this reason, there is a compelling need to unveil molecular details of selectivity determinants beyond the Gα C-terminus which were so far only theoretically predicted for example by an evolutionary study[6] but still lack experimental validation.

Based on our previous finding that in permeabilized cells Gq proteins bind in a more stable manner than Gi/o proteins to muscarinic M₃ receptors (M₃R)[4], this study aimed at identifying the key structures in these Gα subunits which crucially affect the coupling properties. For this purpose, various Gq-based and Go-based Gα chimeras were created by systematically swapping amino acids located at the contact surface with the GPCR. FRET measurements under nucleotide depleted conditions enabled assessment of the binding stability of Gα chimeras at the Gq-coupled M₃ and histaminic H₁ receptor (H₁R) by comparing the dissociation kinetics upon agonist withdrawal. Moreover, the activation potencies of Gα chimeras were determined in intact cells by evaluating concentration-response curves of the interaction between GRK2 and Gβγ subunits released from activated Gα subunits as a measure for physiologically relevant coupling efficiencies.

For M₃R and H₁R, our assays demonstrate that binding and activation characteristics of critical Gα structures extending beyond the C-terminus can be transferred onto Gα chimeras that are based on a different G protein class. Furthermore, this study revealed their distinguishable impact on coupling specificity at these consecutive events which may be receptor-specific.

## Results

**Distinct stabilities of M₃ receptor-G protein complexes.** Interactions between YFP-labeled receptors and CFP-labeled Gγ₂ subunits interpreted as a G protein binding to the receptor were measured by means of FRET in single permeabilized HEK293T cells as illustrated in Fig. 1a. After washout of the nucleotides, agonist stimulation resulted in forming a stable ternary complex that was reversible upon washout of the agonist or by application of GTPγS (Fig. 1b, Supplementary Fig. 1a). In response to agonist withdrawal, M₃ receptors formed more stable complexes with Gq proteins in comparison to Go proteins, whereas endogenous Gα subunits (empty pcDNA3 vectors transfected instead of Gα subunits) did not lead to noticeable FRET signals (Fig. 1b). Receptor-G protein dissociation kinetics in the absence of nucleotides were quantified by calculating the area under the curve (AUC) of normalized traces to allow for analysis of different types of decays (Fig. 1b, see magnification). Furthermore, the examination of the second decay ensured the entire depletion of potential remaining nucleotides as they were continuously purged away by the perfusion. As the evaluation of kinetics rather than amplitudes should be independent of the relative Gα expression, the expression levels were not determined as long as clear FRET signals could be detected. The significant differences in the dissociation kinetics of M₃R-Gq versus M₃R-Go complexes were the starting point for this study in which we studied the influence of potential G protein interaction sites on coupling specificity. Based on the results from many different studies which showed that the C-terminus (α5 helix) of the Gα subunit is important for Gα-subtype-selective G protein activation, we assumed that it is also crucial for the binding stability at the receptor[25,28]. Therefore, we investigated the influence of various segments of the α5 helix on the stability of M₃R-G protein complexes by generating C-terminally modified chimeric Gα subunits.

Chimeric Go subunits were cloned with the aim of receiving Gq-like properties (Fig. 1c). Schemes indicate that in GoqC11 the last 11 (half of the α5 helix) and in GoqC22 the last 22 amino acids (full α5 helix) were replaced by corresponding amino acids of Gαq, whereas GoqC22-11 contains the inner C-terminal 11 amino acids of Gq while the outer ones still belong to Go. After the withdrawal of acetylcholine (ACh), there was no significant difference between the AUC of Go and GoqC11, contrary to expectations that the outermost C-terminus is the crucial binding partner of the receptor as implied by recently published GPCR-G protein structures[7–20]. However, constructs containing the inner α5 helix (GoqC22-11) or the full α5 helix of Gαq (GoqC22) dissociated slower than Go from the M₃R, suggesting that the N-terminal half of the α5 helix contributed far more to the stability of the M₃R-G protein complex than the C-terminal half (Fig. 1c).

Conversely, Gq-based chimeras (Fig. 1d) bearing parts of the C-terminal helix of Gαo were cloned analogously to Fig. 1c. Containing the C-terminal part of the α5 helix of Gαo, GqoC11 dissociated only slightly faster than Gq from the M₃R, resulting in a reduced AUC of less than 5% (Fig. 1d). In contrast, the exchange of the N-terminal half of the α5 helix in GqoC22-11 and the full exchange in GqoC22 completely prevented binding (Supplementary Fig. 1b) even though the proper expression could be confirmed by immunoblots (Supplementary Fig. 1c, d).

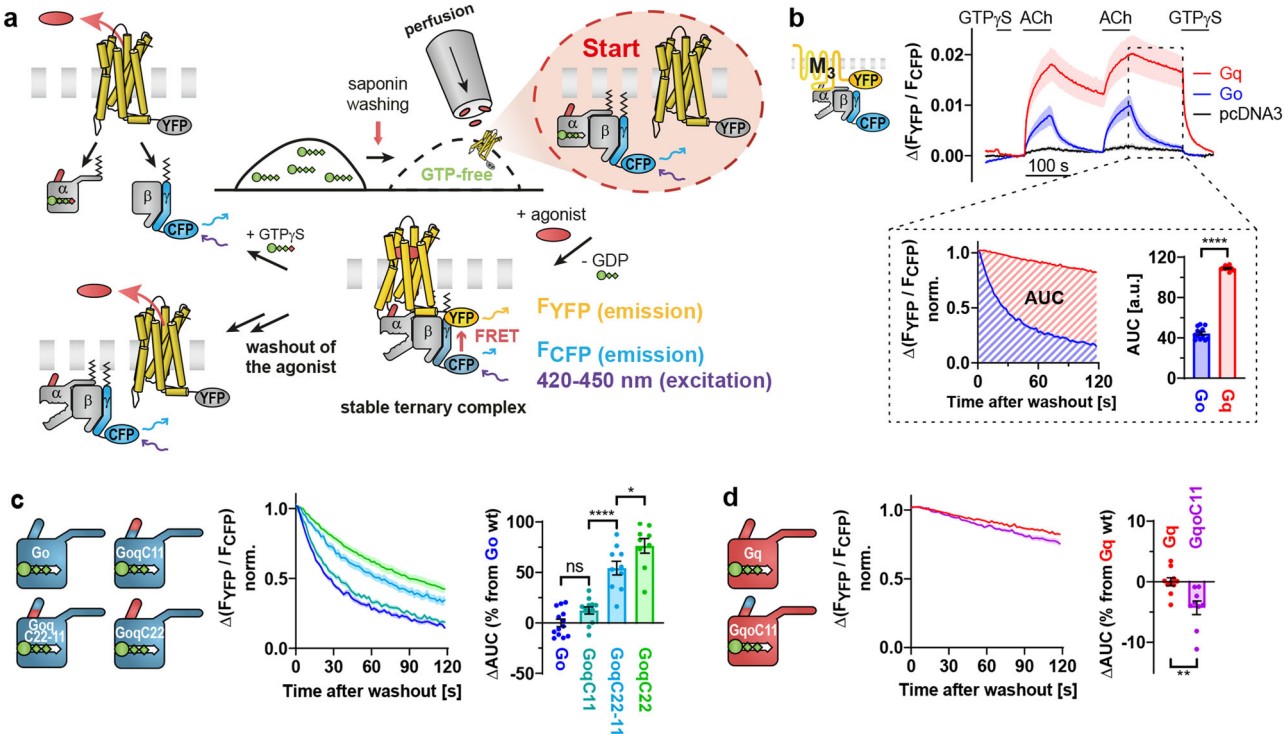

**Fig. 1 Distinct stabilities of M3 receptor-G protein complexes. a–d** G protein binding measurements were performed in single permeabilized HEK293T cells so that nucleotides could be washed out. As depicted in the first scheme, individual cells transfected with C-terminally mCitrine-labeled M$_3$ receptors, various Gα subunits, Gβ$_1$ and N-terminally mTurquoise2-labeled Gγ$_2$ subunits were illuminated at 420–450 nm to excite the CFP-variant, while simultaneously recording the YFP/CFP emission ratio. Cells were continuously superfused with buffer or buffer containing different solutions, as indicated by the bars above the traces in **b**. After short application of 1 μM GTPγS, the cells were stimulated twice with 10 μM acetylcholine (ACh) for 1 min followed by a 2 min washout phase and a final GTPγS application to induce full G protein dissociation from the M$_3$R. To correct for photobleaching, the baselines were subtracted at the level of GTPγS. All data points are represented as means ± SEM. **b** Averaged traces for Gq (red, n = 10), Go (blue, n = 13) and pcDNA3 (black, n = 12, empty vector instead of a transfected Gα subunit) binding to M$_3$R illustrate the time protocol and the absolute amplitudes. Dissociation kinetics after agonist removal were quantified by calculating the area under the curve (AUC) of traces normalized to the amplitude of the second peak as indicated in the magnified insert. The schemes of chimeric Gα subunits in **c** and **d** illustrate the C-terminal amino acids switched between Gq and Go. The experiments were performed in analogy to **b**. Traces are colored as indicated in the corresponding bar graphs (Go = blue, GoqC11 = turquoise, GoqC22-11 = light blue, GoqC22 = green, Gq = red, GqoC11 = purple). Dissociation kinetics of M$_3$R-G protein complexes were quantified by determining the AUC as described in **b** and plotted in the bar graphs as relative changes of the AUC from the respective wild-type G protein. (**c**: Go; n = 13 duplicated from **b**, GoqC11; n = 12, GoqC22-11; n = 10 and GoqC22; n = 9. **d**: Gq; n = 10 duplicated from **b** and GqoC11; n = 9). Statistical analyses were performed using a one-way ANOVA followed by Tukey's posttest (**c**, *P < 0.05, ****P < 0.0001, ns P ≥ 0.05) or a one-tailed t-test (**d**, **P < 0.01).

Therefore, the N-terminal half of the α5 helix seems to play a critical role in binding to M$_3$ receptors. Absolute AUC values of all binding measurements are provided in Supplementary Fig. 2.

**Activation of C-terminally modified G proteins by M$_3$ receptors.** Based on the previously cloned Gα chimeras, we intended to investigate how the stability of M$_3$R-G protein complexes will subsequently be translated into physiologically relevant G protein activation in intact cells. For this reason, FRET-based measurements were performed by analyzing the recruitment of YFP-labeled GRK2 by CFP-labeled βγ subunits which had to be dissociated from activated Gα subunits. Thus, the Gα activation could be investigated indirectly while stimulating the M$_3$R with increasing concentrations of carbachol (representative cells; Supplementary Fig. 3a, b). For quantification, individually calculated EC50 values related to the native Gα subunit were depicted in horizontal bars. As a result, the C-terminal α5 helix of Gαq in GoqC11 significantly increased the potency over Go (Fig. 2a) contrary to a fairly unaffected binding stability at the M$_3$R which remained at the level of Go (Fig. 1c). With the fully exchanged α5 helix, GoqC22 displayed the most left-shifted curve, whereas GoqC22-11 containing only the N-terminal half of

the α5 helix of Gαq showed up between Go and GoqC22, similar to GoqC11. This suggests that both the outermost 11 and the inner 11 amino acids of the α5 helix might contribute equally to increased activation potencies. Even though endogenous G proteins are also activated in this assay, which could not be prevented by a pertussis toxin (PTX) pretreatment (Supplementary Fig. 3c), chimeric Gα subunits showed significantly higher FRET signals at the maximum carbachol concentration than cells transfected with empty vectors (pcDNA3) instead of Gα subunits (vertical bar graphs; Fig. 2a, b).

Conversely, by replacing the C-terminal α5 helix in Gαq with Gαo, GqoC11 substantially lowered the activation potency nearly to the level of Go (Fig. 2b) although the Gq-like binding stability at the M$_3$R was still retained (Fig. 1d). Furthermore, GqoC22 could not be activated at all (Supplementary Fig. 3d) which may be explained by the already interrupted binding (Supplementary Fig. 1b).

To prove if the rather indirect activation assay provided reliable results, we decided to directly assess the G protein activation as well by analyzing decreasing YFP/CFP emission ratios upon activation of YFP-labeled Gα and CFP-labeled Gγ$_2$ subunits (Supplementary Fig. 4). In this assay, the potency shifts of

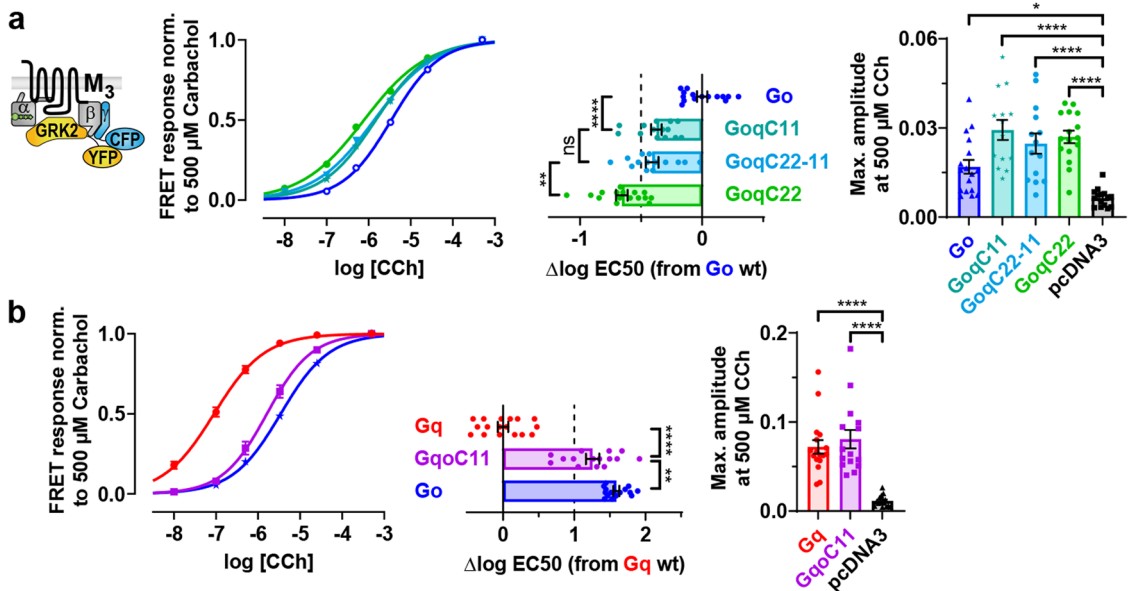

**Fig. 2 Activation of C-terminally modified G proteins by M3 receptors. a–b** G protein activation measurements were performed in intact HEK293T cells illuminated at 420–450 nm to excite the CFP-variant while simultaneously recording the YFP/CFP emission ratio. Cells were continuously superfused with buffer or buffer containing carbachol (CCh) with increasing concentrations up to a maximum of 500 μM. Concentration-response curves were fitted by a variable slope, the bottom fixed to 0 and the top to 1. Individually calculated EC50 values were plotted in horizontal bars as relative changes from the averaged EC50 values of the native G proteins. All data points are represented as means ± SEM (error bars in the concentration-response curves are not visible if they were smaller than the size of the symbol) and colored as indicated in the corresponding bar graphs (Go = blue, GoqC11 = turquoise, GoqC22-11 = light blue, GoqC22 = green, pcDNA3 = black, Gq = red, GqoC11 = purple). G protein activation was measured indirectly recruiting GRK2 by βγ-subunits with cells transfected with M3 receptors, various Gα subunits, Gβ1, N-terminally mTurquoise2-labeled Gγ2 subunits and a C-terminally mCitrine-labeled GRK2. Vertical bar graphs depict the maximum amplitudes compared to an empty vector (pcDNA3) instead of a transfected Gα subunit. (**a** Go; n = 15, GoqC11; n = 14, GoqC22-11; n = 14, GoqC22; n = 16 and pcDNA3; n = 15. **b** Gq; n = 17, GqoC11; n = 15, pcDNA3; n = 16 and Go; n = 15 duplicated from **a**). Statistical analyses were performed using one-way ANOVAs followed by Tukey's posttests for the EC50 values or Dunnett's posttests to compare against pcDNA3 for the maximum amplitudes (**a–b**, *P < 0.05, **P < 0.01, ****P < 0.0001, ns if P ≥ 0.05).

GoqC11 and GqoC11 in relation to the respective native Gα subunit correlated well with the ones measured indirectly (Fig. 2a, b). However, the distance between the curves of Gq and Go were smaller in the direct measurement which might be attributed to differently located fluorophore insertions in Go-YFP versus Gq-YFP that seemed to alter their potency and presumably impeded their direct comparison. For this reason, the indirect activation assay with the more wild-type-like Gα subunit seemed to be more reasonable.

**Examination of chimeric Go subunits achieving Gq-like binding and activation properties with M3R.** As the α5 helix alone only partially contributed to binding at the M3R, we investigated new Gα structures which were previously described to interact with the receptor[22,30]. In our experiments, several of these structures were able to transfer Gq-like coupling properties onto Go-based chimeras. As schematically depicted in Fig. 3a, amino acids of Go were replaced by those of Gq from either the whole N-terminus including the αN/β1 hinge (GoqN), the connection of the β2/β3 sheets (Goq2), or the loop between α4 helix and β6 sheet (Goq4). In addition, Go-based chimeras were also combined with various lengths of exchanged α5 helices. Details of the swapped amino acids are illustrated in Supplementary Fig. 5a, b. In Fig. 3b, the chimeric sites are depicted by different views of the Gα surface relative to the receptor using the most related structure of the M1R-G11 complex (pdb: 6oij)[19]. The magnification highlights the intracellular loop 2 (ICL2) of the receptor which fits in a cleft formed by the αN/β1 hinge, the β2/β3 loop, and the N-terminal half of the α5 helix of the Gα subunit. Unfortunately, the ICL3 is not resolved in the structure but might

still interact with the α4/β6-loop as the ICL3 is quite big for muscarinic receptors.

Remarkably, by replacing the N-terminus in Gαo with Gαq, GoqN chimeric constructs dissociated considerably slower than Go from the M3R (Fig. 3c). Especially double chimeras exhibited a substantially enhanced complex stability so that GoqN + C22 containing both the N-terminal and C-terminal helices of Gαq achieved almost a Gq-like binding stability. Moreover, with the β2/β3 loop of Gαq, Goq2 and Goq2 + C11 dissociated more slowly from the M3R in a similar pattern (Fig. 3d) even though Goq2 + C22 did not further increase the AUC.

Activation measurements of GoqN constructs (Fig. 3e) revealed a similar trend of increased potencies according to elevated AUCs in the binding experiments. This increase in potency was accompanied by an apparent flattening of the concentration-response curves, specifically for GoqN + C22 and GoqN + C11, a phenomenon which could not be attributed to the interference of endogenous Gi/o proteins as pertussis toxin pretreatment did not steepen the curve of GoqN + C22 (Supplementary Fig. 6). For Goq2 chimeras, we observed considerably higher activation potencies than for Go (Fig. 3f) which were almost as high as for the GoqN + C11 double chimera (Fig. 3e). However, the EC50 could not be further enhanced by additionally exchanged C-termini in Goq2 + C11 or Goq2 + C22. Maximum amplitudes for all activation measurements in comparison to pcDNA3 are provided in Supplementary Fig. 7.

To narrow down the region of the N-terminal exchange important for the stability of M3R-GoqN-complexes, we generated additional chimeric Gα subunits (Goq1, Goq1 + C11, and

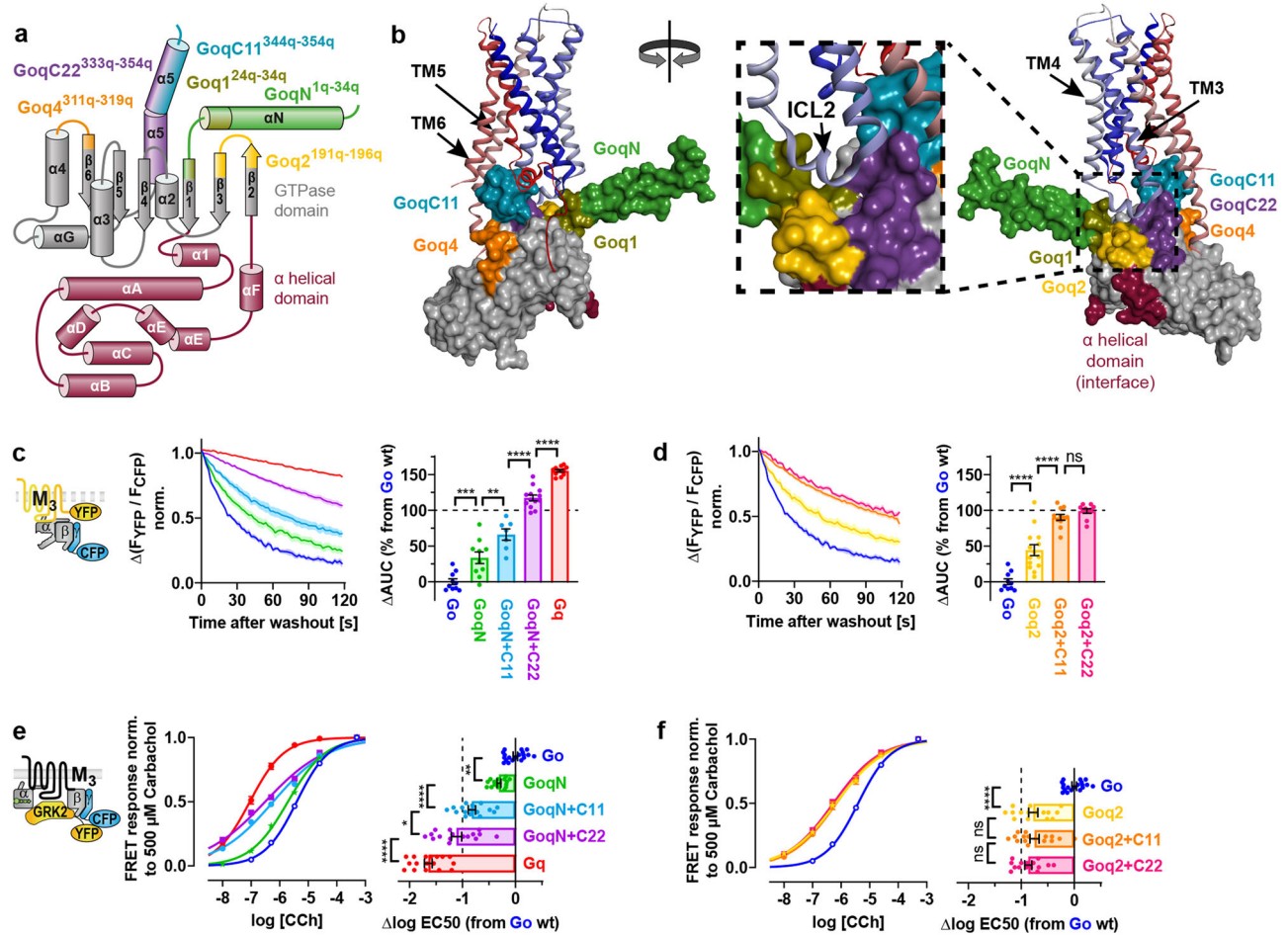

**Fig. 3 Examination of chimeric Go subunits achieving Gq-like binding and activation properties with M3R. a** Colored segments in the schematic G protein structure illustrate amino acids in Go replaced by the corresponding ones from Gq in various chimeric Goq subunits (details in Supplementary Fig. 5a, b). **b** Utilizing the $M_1R$-$G_{11}$ complex structure (pdb: 6oij)[19], chimeric structures are depicted by different views of the $G_{11}$ surface in relation to the $M_1R$ (N-terminus: dark-blue to C-terminus: red). The C-terminal end of the αN helix (aa 30–35), being part of Goq1 and GoqN is not resolved in the structure. **c**–**d** G protein binding experiments were performed in analogy to Fig. 1b. Dissociation kinetics of $M_3R$-G protein complexes were quantified by determining the AUC as described in Fig. 1b and plotted in the bar graphs as relative changes of the AUC from Go. (**c** Go; $n = 10$, GoqN; $n = 10$, GoqN + C11; $n = 7$, GoqN + C22; $n = 13$, and Gq; $n = 12$. **d** Goq2; $n = 14$, Goq2 + C11; $n = 11$, Goq2 + C22; $n = 10$, and Go; $n = 10$ duplicated from **c**). **e**–**f** Indirect G protein activation measurements were performed in analogy to Fig. 2a, b. Individually calculated EC50 values were plotted in horizontal bars as relative changes from the averaged EC50 of Go. (**e** Go; $n = 19$, GoqN; $n = 16$, GoqN + C11; $n = 16$, GoqN + C22; $n = 16$ and Gq; $n = 17$ duplicated from Fig. 2b. **f** Goq2; $n = 15$, Goq2 + C11; $n = 16$, Goq2 + C22; $n = 15$ and Go; $n = 19$ duplicated from **e**). All data points are represented as means ± SEM (error bars in the concentration-response curves are not visible if they were smaller than the size of the symbol) and colored as indicated in the corresponding bar graphs (Go = blue, GoqN = green, GoqN + C11 = light blue, GoqN + C22 = purple, Gq = red, Goq2 = yellow, Goq2 + C11 = orange, Goq2 + C22 = pink). Statistical analyses were performed using one-way ANOVAs followed by Tukey's posttests (**c**–**f**, \*$P < 0.05$, \*\*$P < 0.01$, \*\*\*$P < 0.001$, \*\*\*\*$P < 0.0001$, ns if $P \geq 0.05$).

Goq1 + C22) carrying only 11 amino acids of the αN/β1 hinge of Gαq instead of the whole N-terminus (Fig. 3a, b). Even though their binding stabilities at the $M_3R$ resembled their GoqN counterparts (Supplementary Fig. 8a), activation potencies were hard to measure due to low amplitudes probably resulting in merged EC50 values with less potent endogenous G proteins (Supplementary Fig. 8b).

The Goq4 chimera which carries the connection between the α4 helix and the N-terminal part of the β6-sheet of Gαq addressing contacts to the ICL3 of the receptor was similar to Go in binding kinetics (Supplementary Fig. 9a) and indistinguishable in activation (Supplementary Fig. 9b). Therefore, this structure plays no vital role in the generation of G protein specificity for the $M_3R$.

Our results so far show that not only the N-terminal half of the α5 helix of Gαq increases both the stability in $M_3R$-complexes, as

well as coupling efficiency but also the αN helix with the connection to the β1 sheet and parts of β2 and β3 sheet including their linker induce better $M_3R$ binding and coupling. Moreover, combinations of these Gq structures in addition to the α5 helix of Gαq almost lead to a complete transfer of Gq-like coupling properties onto Go-based double chimeras.

**Chimeric Gq subunits retain their stability in $M_3$ receptor complexes while activation is impaired.** To check if coupling properties can also be switched the other way around, Gq-based chimeras comprising structures of Gαo were cloned in analogy to Goq chimeras (Fig. 3a, b) and tested for their binding to $M_3R$ and coupling efficiency. However, replacing the N-terminus in Gαq by Gαo revealed indistinguishable binding kinetics between GqoN and Gq (Fig. 4a), whereas the GqoN + C11 double chimera

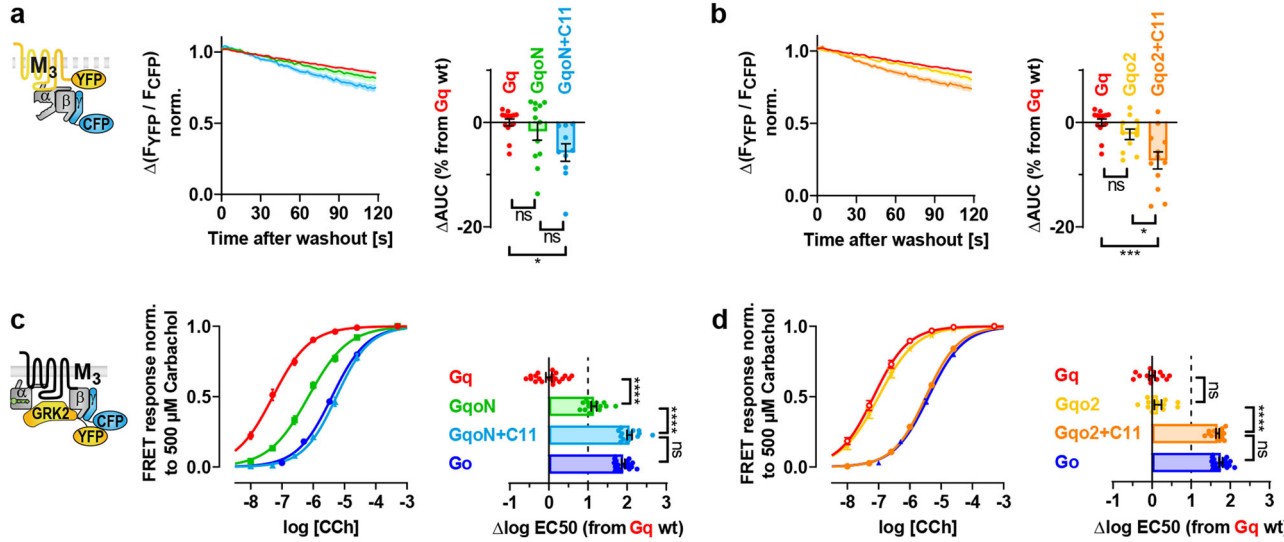

**Fig. 4 Chimeric Gq subunits retain their stability in M3 receptor complexes while activation is impaired. a–b** G protein binding experiments of chimeric Gqo proteins containing conversely exchanged structures as illustrated in Fig. 3a, b were conducted similarly to Fig. 1b. Dissociation kinetics of $M_3R$-G protein complexes were quantified by determining the AUC as described in Fig. 1b and plotted in the bar graphs as relative changes of the AUC from Gq. (**a** Gq; n = 14, GqoN; n = 12 and GqoN + C11; n = 10. **b** Gqo2; n = 11, Gqo2 + C11; n = 12 and Gq; n = 14 duplicated from **a**). **c–d** Indirect G protein activation measurements of Gqo constructs were performed in analogy to Fig. 2a, b. Individually calculated EC50 values were plotted in horizontal bars as relative changes from the averaged EC50 of Gq. (**c** Gq; n = 20, GqoN; n = 14 GqoN + C11; n = 15, and Go; n = 19 duplicated from Fig. 3e. **d** Gqo2; n = 12, Gqo2 + C11; n = 15, Gq; n = 14 and Go; n = 19 duplicated from Fig. 3e). All data points are represented as means ± SEM (error bars in the concentration-response curves are not visible if they were smaller than the size of the symbol) and colored as indicated in the corresponding bar graphs (Gq = red, GqoN = green, GqoN + C11 = light blue, Gqo2 = yellow, Gqo2 + C11 = orange, Go = blue). Statistical analyses were performed using one-way ANOVAs followed by Tukey's posttests (**a–d**, *P < 0.1, ***P < 0.001, ****P < 0.0001, ns if P ≥ 0.05).

dissociated only less than 10% faster than Gq from the $M_3R$. Similar results are true for exchanges of the β2/β3 loop in Gqo2 and Gqo2 + C11, which could mainly retain the Gq-like binding properties (Fig. 4b).

By contrast, the activation potencies of GqoN constructs were considerably lower than for Gq. The curve of GqoN (Fig. 4c) was strongly right-shifted and GqoN + C11 even further, indistinguishable from Go. In contrast, Gqo2 (Fig. 4d) did not lose potency, but the Gqo2 + C11 double chimera was right-shifted to the level of Go as well. However, the low potencies of the double chimeras were not surprising as the sole exchange of the C-terminal α5 helix in GqoC11 led already to a potency loss close to Go (Fig. 2b).

The combinations GqoN + C22 and Gqo2 + C22 could neither be bound (Supplementary Fig. 10a) nor activated (Supplementary Fig. 10b) by the $M_3R$ as previously shown with the GqoC22 single chimera (Supplementary Fig. 1b, c). Furthermore, Gqo4 comprising the α4/β6 loop of Gαo behaved identically to Gq in binding (Supplementary Fig. 11a) and activation assays (Supplementary Fig. 11b) which are in line with the Goq4 counterpart that equaled Go (Supplementary Fig. 9a, b).

Overall, the binding stability of Gq-based chimeras at the $M_3R$ can barely be lowered by replacing the N-terminus, the β2/β3 loop or additionally the C-terminal α5 helix by Gαo, whereas Go-based chimeras are able to gain distinctly enhanced binding properties if these structures are exchanged by Gαq (Fig. 3c, d). Even though the Gq-backbone ensures stable binding of Gqo chimeras to $M_3R$, they are subsequently selectively activated by the $M_3R$.

**Binding and activation measurements of chimeric Goq and Gqo subunits with $H_1$ receptors.** After the identification of regions on Gαq and Gαo which are important for binding and coupling to $M_3$ receptors, we wanted to test if these Gα structures

are recognized by other Gq-coupled receptors as well. Therefore, we repeated the measurements with $H_1$ receptors ($H_1R$), which—even though closely related to $M_3R$—have most probably independently developed the specificity of G protein coupling during evolution[6]. Similar to the $M_3R$ (Fig. 1c), the replacement of the whole α5 helix in Gαo by Gαq slowed the dissociation of GoqC22 from the $H_1R$ (Fig. 5a). However, different from the $M_3R$, the $H_1R$-G protein complex stability of GoqC11 was slightly increased over Go, whereas the stability of GoqC22-11 containing only the N-terminal half of the α5 helix of Gαq was not increased to a significant extent (Fig. 5a).

The relevance of the N-terminal helix of Gq for binding to $H_1R$ was also confirmed (Fig. 5b). The dissociation of GoqN chimeras from $H_1R$ was even slower than from $M_3R$, so that the additional C-terminal exchange in GoqN + C11 resulted in complex stability close to Gq, whereas GoqN + C22 exhibited similar $H_1R$-G protein complex stabilities. The β2/β3 loop of Gαq included in Goq2 constructs also improved the stability over Go in $H_1R$ complexes (Fig. 5c) in good agreement with $M_3R$ measurements (Fig. 3d). Remarkably, the double chimera Goq2 + C11 even reached the AUC of Gq, whereas for Goq2 + C22 the AUC was significantly lower. With the $M_3R$ the binding of Goq2 + C22 was not prolonged over Goq2 + C11 either, but also not disturbed. Taken together, chimeric Gq structures of the β2/β3 loop in addition to the N-terminus and C-terminus seemed to crucially enhance the binding stability of Goq chimeras at both the $M_3R$ and $H_1R$.

For Gq-based chimeras, there was a noticeable difference between both receptors, as GqoC11 dissociated distinctly faster than Gq from the $H_1R$ (Fig. 5d) whereas the difference was less pronounced for the $M_3R$ (Fig. 1d). Therefore, the outermost 11 amino acids of the Gα C-terminus seemed to be more important for the stability of $H_1R$ complexes than for $M_3R$ complexes. However, both receptors have in common that GqoC22 did not

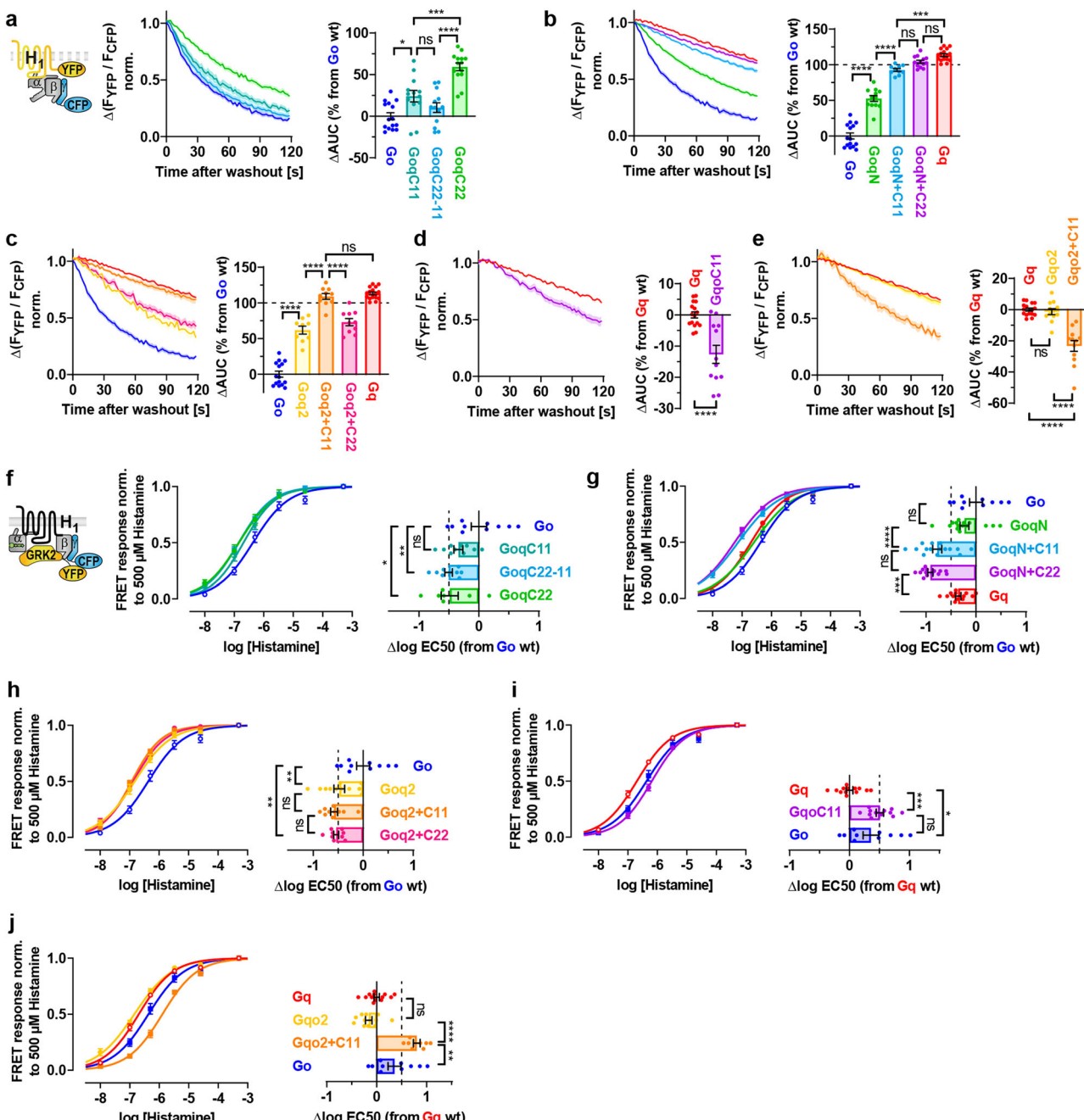

**Fig. 5 Binding and activation measurements of chimeric Goq and Gqo subunits with H1 receptors. a–e** G protein binding experiments of chimeric Goq and Gqo subunits with H$_1$ receptors were performed in analogy to Fig. 1b but transfected with C-terminally mCitrine-labeled H$_1$R instead of mCit-labeled M$_3$R and stimulated with 100 μM histamine instead of 10 μM ACh. Dissociation kinetics of H$_1$R-G protein complexes were quantified by determining the AUC as described in Fig. 1b and plotted in the bar graphs as relative changes of the AUC from Go in **a–c** and from Gq in **d–e**. (**a** Go; $n = 15$, GoqC11; $n = 13$, GoqC22-11; $n = 13$ and GoqC22; $n = 14$. **b** GoqN; $n = 12$, GoqN + C11; $n = 10$, GoqN + C22; $n = 13$, Gq; $n = 16$ and Go; $n = 15$ duplicated from **a**. **c** Goq2; $n = 9$, Goq2 + C11; $n = 10$, Goq2 + C22; $n = 10$, Go; $n = 15$ duplicated from **a** and Gq; $n = 16$ duplicated from **b**. **d** GqoC11; $n = 12$ and Gq; $n = 16$ duplicated from **b**. **e** Gqo2; $n = 12$, Gqo2 + C11; $n = 12$ and Gq; $n = 16$ duplicated from **b**). **f–j** Indirect G protein activation measurements were performed in analogy to Fig. 2a, b but transfected with H$_1$R instead of M$_3$R and stimulated with the maximum of 500 μM histamine instead of 500 μM CCh. Individually calculated EC50 values were plotted in horizontal bars as relative changes from the averaged EC50 values of Go in **f–h** and of Gq in **i–j** (**f** Go; $n = 10$, GoqC11; $n = 10$, GoqC22-11; $n = 9$ and GoqC22; $n = 7$. **g** GoqN; $n = 15$, GoqN + C11; $n = 15$, GoqN + C22; $n = 14$, Gq; $n = 13$ and Go; $n = 10$ duplicated from **f**. **h** Goq2; $n = 11$, Goq2 + C11; $n = 10$, Goq2 + C22; $n = 9$ and Go; $n = 10$ duplicated from **f**. **i** GqoC11; $n = 12$, Gq; $n = 13$ duplicated from **g** and Go; $n = 10$ duplicated from **f**. **j** Gqo2; $n = 11$, Gqo2 + C11; $n = 8$, Gq; $n = 13$ duplicated from **g** and Go; $n = 10$ duplicated from **f**). All data points are represented as means ± SEM and colored as indicated in the corresponding bar graphs. Statistical analyses were performed using one-way ANOVAs followed by Tukey's posttests (**a–j**, *P < 0.1, **P < 0.01, ***P < 0.001, ****P < 0.0001, ns if P ≥ 0.05).

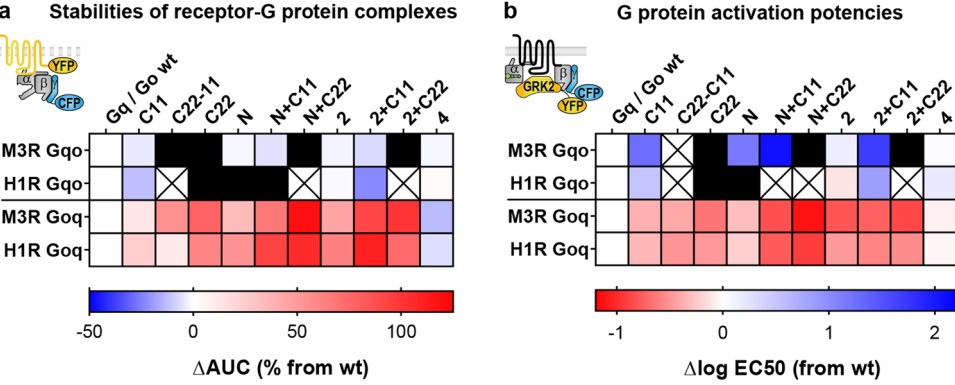

**Fig. 6 Side-by-side comparison of coupling properties between M3R and H1R. a–b** Summary of binding and activation experiments of chimeric G proteins measured in Figs. 1–5 is depicted side by side in heat maps. Rows describe investigated receptors and G protein classes from which the chimeric constructs originate. Columns represent segments where amino acids were replaced by the respective other G protein class. Their sites are illustrated in Fig. 3a, b. All data points are represented as means. Black areas indicate constructs showing no specific signal over endogenous G proteins. Crossed areas mean that no data were acquired for that construct. **a** Colored areas show the ΔAUC relative to the native G protein depicted in red for a more stable or in blue for a less stable binding in receptor-G protein complexes compared to wild-type G proteins (white). **b** Tiles represent relative EC50 values of indirect G protein activation measurements shown in red for gains and in blue for losses of activation potency compared to wild-type G proteins (white).

bind at all (Supplementary Fig. 12a and Fig. 1b). Furthermore, Gqo2 in which the β2/β3 loop in Gαq was replaced by Gαo still exhibited Gq-like kinetics (Fig. 5e) in line with the M₃R (Fig. 4b). The faster dissociation of Gqo2 + C11 from the H₁R compared to the M₃R was consistent with the influence of the Go C-terminus in GqoC11 leading to a quite fast dissociation from the H₁R as mentioned before. Therefore, the exchange of the β2/β3 loop accelerated the dissociation only slightly more in the combined Gqo2 + C11 compared to GqoC11, virtually the same as with the M₃R.

The most obvious difference between H₁R and M₃R was exerted by the exchange of the N-terminus as the binding of GqoN and also GqoN + C11 to the H₁R was completely impaired (Supplementary Fig. 12b). In addition, no activation was observed (Supplementary Fig. 12c), further suggesting that the N-terminus of Gαo was detrimental for the interaction of Gq-based chimeras with the H₁R. In contrast, GqoN constructs interacted well with the M₃R (Fig. 4a) and were still activated (Fig. 4c).

For G protein activation measurements carried out with the H₁R instead of the M₃R, the differences in the concentration-response curves for specific G protein chimeras appeared to be less pronounced but still show a chimera-specific pattern. By swapping parts of the α5 helix in GoqC constructs, most activation potencies were shifted to a similar extent with H₁R (Fig. 5f) and M₃R (Fig. 2a). Even though the maximum amplitude of GoqC22 was just below the significance limit compared to pcDNA3 ($P = 0.08$; Supplementary Fig. 7), it was analyzed because of a pronounced left-shifted curve. However, the small amplitude may explain the slightly lower potency gain compared to the M₃R due to competition with less potent endogenous G proteins. Remarkably, the activation of Go-based chimeras containing the N-terminal and C-terminal helices of Gq (GoqN + C11 and GoqN + C22) even exceeded the activation potency of Gq (Fig. 5g). This suggests that in contrast to the M₃R, the H₁R may even favor a structure in the Go backbone while potencies were still further elevated by the N-terminal and C-terminal exchange similar to the M₃R. The concentration-response curves of all Goq2-like chimeras were equally left-shifted (Fig. 5h) and thus also independent of a specific C-terminus similar to the M₃R (Fig. 3f).

Finally, the activation of Gq-based chimeras by the H₁R was investigated. The curve of GqoC11 appeared approximately at the level of Go (Fig. 5i) and thus hardly further right-shifted than

activated by the M₃R (Fig. 2b). In parallel, Gqo2 (Fig. 5j) revealed an unaltered potency at the level of Gq similar to the activation by the M₃R. However, the curve of Gqo2 + GqoC11 activated by the H₁R appeared even further right-shifted than Go, whereas the M₃R always exhibited the weakest EC50 with Go (Fig. 4d). This finding might be explained by a Go backbone that is preferred for activation by the H₁R in line with the previously mentioned GoqN + C11 and GoqN + C22 double chimeras that could be activated even more easily than Gq (Fig. 5g).

The exchanges of the α4/β6 loop in Goq4 (Supplementary Fig. 13a, b) and also Gqo4 (Supplementary Fig. 13c, d) did not alter complex stabilities or activation potencies compared to the native Gα subunits similar to the M₃R findings.

In summary, critical structures like the β2/β3 loop together with the N-terminus and C-terminus of the Gα subunit are essential for proper binding and activation by the H₁R, as well as by the M₃R. However, selectivity barriers for each receptor can interfere at different time points during the coupling process, depending on the investigated structure. For instance, the exchanged C-terminal α5 helix in GqoC11 clearly impairs the binding stability at the H₁R, whereas for M₃R the subsequent activation of GqoC11 is considerably stronger compromised than the only slightly altered binding. The same is true for the exchanged N-terminus in GqoN which prevents binding to H₁R whereas for the M₃R only the subsequent activation is negatively affected. Nevertheless, most of the Gα chimeras are already specifically selected at the level of binding by both receptors.

**Side-by-side comparison of coupling properties between M₃R and H₁R.** For a side-by-side comparison, we generated heat maps that include all Gα chimeras measured with M₃R and H₁R. Receptor-G protein complex stabilities in permeabilized cells are represented by the AUC relative to the native Gα subunit (Fig. 6a). In summary, there were only minor differences between the M₃R and the H₁R in the binding of Gα chimeras except for GqoN which did not bind to H₁R (depicted as a black panel) but did bind to M₃R, confirming the crucial role of the N-terminus for ternary complex formation and stability. Therefore, both Gq-coupled receptors seem to selectively recognize Gα subunits by their N-terminus, β2/β3 loop, and C-terminus.

G protein activation measurements in intact cells are represented by EC50 values relative to the native Gα subunit (Fig. 6b). Overall,

potency changes of Gα chimeras correlate well with the respective binding measurements (Fig. 6a) except for Gqo chimeras activated by M$_3$R which mostly revealed remarkably strong potency losses contrary to unaltered M$_3$R complex stabilities. This suggests that specificity for Gq-based chimeras might only be exerted at subsequent steps of the coupling process for the M$_3$R.

## Discussion

Collectively, this study unveils several key structures beyond the C-terminus in the Gα subunit which are crucially involved in the process of selective binding and coupling to Gq-coupled receptors. Moreover, comprehensive FRET measurements allowed for the differential analysis of distinct coupling events such as G protein binding and activation and their influence on specific recognition of multiple Gα chimeras by M$_3$ and H$_1$ receptors. For this purpose, we assessed the binding of nucleotide-free Gα subunits to agonist-occupied receptors by analyzing dissociation kinetics of stably formed complexes to quantify otherwise static interactions in recently published structures. However, early association events that might influence coupling specificity already before GDP release could not be assessed. In a subsequent step, we investigated how these complex stabilities were ultimately translated into physiologically relevant G protein activation reflected by the release of activated Gα subunits from Gβγ subunits whose interaction with GRK2 was then analyzed.

In our study, the comparisons of Gα chimeras with their related wild-type subunits revealed that under nucleotide-free conditions M$_3$ receptors bind quite promiscuously to Go-based and Gq-based chimeras regardless of whether the distal 11 amino acids of the C-terminus were replaced by the other G protein class (in GoqC11 and GqoC11). These findings further extend the results of a previous study that lacked kinetic data to provide our measurements[29]. In contrast, the binding of GoqC11 to H$_1$R was slightly enhanced, whereas GqoC11 was less stable bound at the H$_1$R. However, this receptor-specific difference may not be unexpected because of a particularly large interface that is shared between the helical bundle of the receptor and the exposed end of the Gα C-terminal helix. Remarkably, even though the binding to the M$_3$R was predominantly unaffected, our evidence suggests that the outermost Gα C-terminus may be critical to its subsequent activation as the activation potencies of GoqC11 and GqoC11 were substantially altered compared to Go or Gq, respectively. These results correlate well with older functional studies which showed that the amount of second messenger generation depends on the distal C-terminal helix of Gα[25,28]. Overall, our binding and activation experiments revealed for the M$_3$R that specificity for the 11 most C-terminal amino acids of Gα seems to be primarily exerted after the binding had already taken place. As this might be receptor-specific, there is a need for further investigations of different receptors.

In contrast to these results, the entire 22 amino acid exchange of the α5 helix already enhanced the initial binding of GoqC22 constructs to both receptors to a significant amount which was subsequently also reflected in higher activation potencies. In the opposite case, GqoC22 chimeras did not bind to these receptors and were not activated. As the same was true if only the N-terminal half of the α5 helix was exchanged in the GqoC22-C11 construct, this is the first study, to our knowledge, to experimentally prove that the N-terminal part of the α5 helix may crucially contribute to binding and activation by Gq-coupled receptors.

Furthermore, our study has demonstrated that the N-terminus including the αN/β1-hinge and the β2/β3-loop are critical structures for selective coupling. Together with the N-terminal half of the α5 helix, they form a cavity in the Gα subunit which was previously described to interact with the ICL2 of the receptor[19,23,31]. However, we found that this interface is not only required for the overall binding but is essential for coupling specificity as the stable binding characteristics from Gq subunits could be transferred to Go-based chimeras by the N-terminus (in GoqN) and the β2/β3 loop (in Goq2) and subsequently also led to increased activation potencies with both receptors. In contrast, the opposite chimeric exchanges were not able to disturb the stable binding properties of Gq-based chimeras like GqoN and Gqo2 to M$_3$ receptors, whereas the activation potency of GqoN was clearly reduced. This suggests that for Gq-based constructs binding to M$_3$R and receptor-induced activation are at least a two-step process with distinguishable properties. For the H$_1$R, Gqo2 also exhibited Gq-like properties, whereas GqoN was even prevented initially from being bound. Since the wild-type Go which contains the same N-terminus as GqoN could still couple to the H$_1$R, it might be assumed that different binding modes were achieved at the receptor depending on the respective Gα backbone. Therefore, a recent study proposing a primary coupling mode for canonical signaling and a secondary coupling mode for non-canonical interactions seems to correlate well with our experiments[31]. As exchanges of the N-terminus or the β2/β3 loop together with the C-terminal helix of Gαq in Goq double chimeras exhibited more than just synergistic effects and thus partially achieved Gq-like properties, it is tempting to suggest that they switched from a Go-like to a Gq-like coupling mode leading to additional and stronger interactions. In contrast, Gqo chimeras seem to retain their original Gq-like binding mode with the M$_3$R, until they get activated. For the M$_3$R-induced activation of Go-based double chimeras, such as GoqN+C22 (Fig. 3e), our data show a curve flattening of the concentration-response curve which cannot be seen for the H$_1$R-induced activation (Fig. 5g). This finding is puzzling, as the contribution of endogenous G proteins could be excluded. We have no obvious explanation for this phenomenon and can only speculate that Gαq-binding to a polybasic motif in the C-terminus of M$_3$R could be involved[32] since H$_1$R lacks this motif.

In conclusion, we differentially assessed Gα chimeras for the first time to our knowledge in regard to their ability to be selectively bound and activated by two different Gq-coupled receptors. We identified Gα structures such as the N-terminus including the αN/β1 hinge, together with the β2/β3 loop and the N-terminal and C-terminal parts of the α5 helix to exert major functions for coupling specificity, whereas the α4/β6 loop, which was mentioned in previous studies to also interact with receptors, did not alter binding or activation properties of Gα subunits in our experiments[22,30]. Furthermore, we found that all Gα chimeras containing Gαo as the backbone were already specifically selected for binding by both the M$_3$R and the H$_1$R. In contrast, for some Gq-based chimeras comprising the C-terminal α5 helix of Gαo (GqoC11) or the N-terminus of Gαo (GqoN) selectivity barriers primarily interfered at the subsequent activation for M$_3$R, contrary to H$_1$R. Therefore, we suggest that selectivity mechanisms are specifically fine-tuned for each receptor and occur at different time points during the multi-step coupling process. In the future, it would be reasonable to also investigate GPCRs with different coupling profiles to expand the scope of this project more toward a general mechanism.

## Methods

**Reagents**. DMEM, FBS, penicillin/streptomycin, L-glutamine, and trypsin-EDTA were purchased from Capricorn Scientific, NEBuilder HiFi DNA Assembly kit and Q5 Polymerase from New England Biolabs, Effectene Transfection Reagent from Qiagen, a saponin from AppliChem, poly-L-lysine hydrobromide, GTPγS, acetylcholine, and carbachol from Sigma-Aldrich, histamine from Alfa Aesar and pertussis toxin from Merck.

**Plasmids**. cDNAs for human $H_1R$-WT[33], mouse $G\alpha q$-WT and $G\alpha q$-YFP (YFP inserted between F124 and E125)[34], PTX insensitive (C351I mutation) rat $G\alpha o$ and $G\alpha o$-YFP (YFP inserted between L91 and G92)[35] and human $G\beta_1$-WT[36] were described previously. $M_3R$-WT was obtained from the cDNA Resource Center (www.cDNA.org) and pcDNA3 from Invitrogen. Bovine mTurq2-$G\gamma_2$, human $M_3R$-mCit, and human GRK2-mCit have been cloned by exchanges of the fluorophores in CFP-$G\gamma_2$[36], $M_3R$-YFP[37] and GRK2-YFP[38] by mTurquoise2 or mCitrine, respectively. For $H_1R$-mCit, mCitrine was C-terminally attached to $H_1R$-WT[33] analogously to $M_3R$-YFP[37].

**Cloning of $G\alpha$ chimeras**. $G\alpha$ chimeras were generated by the Gibson Assembly method without insertion of restriction sites using the NEBuilder HiFi DNA Assembly kit and Q5 Polymerase for PCRs according to the manufacturer's protocols[39]. For Gqo chimeras, $G\alpha q$-WT was used as the backbone, and for Goq chimeras $G\alpha o$ C351I. The inserted DNA was obtained from the respective other $G\alpha$ subtypes. All PCR primers were synthesized and final constructs sequenced by Eurofins Genomics. Oligo sequences are provided in Supplementary Table 1.

**Cell culture and transfections**. Experiments were performed in transiently transfected HEK293T cells (a gift from the Lohse laboratory, University of Wuerzburg, likely from ATCC) which were cultured in DMEM (4.5 g L$^{-1}$ glucose) supplemented with 10% FBS, 2 mM L-glutamine, 100 U mL$^{-1}$ penicillin and 0.1 mg mL$^{-1}$ streptomycin at 37 °C in a humidified atmosphere with 5% $CO_2$. Transfections were conducted in 6 cm dishes using Effectene Transfection Reagent according to the manufacturer's protocol. The transfection for the binding assay contained 0.5 μg $M_3R$-mCit or 0.8 μg $H_1R$-mCit, 1.5 μg of the indicated $G\alpha$, 0.5 μg $G\beta_1$, 0.2 μg mTurq2-$G\gamma_2$ and 0.3 μg pcDNA3 (only if $M_3R$-mCit was used to compensate for the lower DNA amount compared to $H_1R$-mCit). For the activation assay, the transfection contained 0.5 μg $M_3R$ or $H_1R$, 1.5 μg of the indicated $G\alpha$, 0.5 μg $G\beta_1$, 0.2 μg mTurq2-$G\gamma_2$ and 0.3 μg GRK2-mCit. 24 h after transfection, cells were plated on 25 mm poly-L-lysine coated coverslips, and experiments were performed 1 d thereafter.

**FRET measurements**. FRET measurements were carried out at room temperature using an inverted microscope (Axiovert 100; Zeiss), equipped with a ×60 oil-immersion objective (PlanApo N ×60/1.45 Oil; Nikon), LED light sources with excitation intensities set to 4% for 440 nm and 10% for 500 nm (pE-100; CoolLED) and a high-performance CCD-camera (Spot Pursuit from Spot Imaging/Diagnostic Instruments). During FRET measurements, CFP was excited between 420 nm and 450 nm by an excitation filter (436/20; Chroma) and a dichroic beam splitter (458LP; Semrock). Fluorescence emission of CFP and YFP was simultaneously collected side-by-side by a second beamsplitter (505LP; Chroma) and two emission filters (CFP: 470/24; Chroma and YFP: 525/39; Semrock). Cells were illuminated by short light flashes of 60 ms with a frequency of 0.5 Hz while they were superfused via a pressure-driven perfusion system (VC3-8xP Series; ALA Scientific Instruments). Data were collected by the VisiView software (Visitron Systems). Corrections for the background fluorescence, the spillover of CFP into the YFP channel, and the direct YFP excitation at 420–450 nm were calculated in Microsoft Excel 2019. The YFP/CFP emission ratio as a measure of FRET was then adjusted for photobleaching by an exponential baseline subtraction with OriginPro 2018 (OriginLab) resulting in the final FRET ratio indicated as $\Delta(F_{YFP}/F_{CFP})$.

**Binding assay in permeabilized cells**. Prior to the experiment, the coverslip with adherent cells was fixed in a microscope chamber and washed once with external buffer (NaCl 137 mM, KCl 5.4 mM, HEPES 10 mM, $CaCl_2$ 2 mM, $MgCl_2$ 1 mM, pH = 7.35). Then it was incubated for 2.5 min with a 0.075% saponin solution in an internal buffer ($K^+$ aspartate 100 mM, KCl 30 mM, NaCl 10 mM, HEPES 10 mM, EGTA 5 mM, $MgCl_2$ 1 mM, pH = 7.35) and afterward washed five times with internal buffer. The permeabilization process was optimized by an adequate saponin concentration and incubation time followed by sufficient washing steps to ensure nucleotide-free conditions as previously demonstrated[4]. Under the microscope, single cells were selected for membrane staining with CFP and YFP and a round shape as an indicator for full permeabilization. During the measurement, cells were superfused with an internal buffer or buffer containing the indicated agonist or GTPγS. Imaging and general data processing were performed as described in the previous section (see "FRET measurements" section). Examination of the 2nd washout phase of the agonist ensured the entire depletion of potential remaining nucleotides as they were continuously purged away by the perfusion. After the baseline was subtracted to 0 at the level of GTPγS application, the FRET ratio $\Delta(F_{YFP}/F_{CFP})$ was normalized to the second peak set to 1. The stability of receptor-G protein complexes was quantified by the AUC (calculated by the trapezoidal method in Microsoft Excel) in order to compare different types of dissociation kinetics (e.g., linear and exponential). The AUC was plotted in relation to the respective native $G\alpha$ subunit (ΔAUC) with Graphpad Prism 8.4.

**Indirect G protein activation assay: GRK2 recruitment by $G\beta\gamma$ subunit**. As an indirect measure of G protein activation, mCitrine-labeled GRK2 is recruited by mTurquoise2-labeled βγ subunits after dissociation from activated $G\alpha$ subunits. Prior to the experiment, adherent cells were washed once with external buffer

(NaCl 137 mM, KCl 5.4 mM, HEPES 10 mM, $CaCl_2$ 2 mM, $MgCl_2$ 1 mM, pH = 7.35). Under the microscope, a group of cells was selected for membrane staining of CFP and bright cytosolic staining of YFP and was continuously superfused during the measurement with external buffer or the consecutive application of buffer containing increasing agonist concentrations up to 500 μM carbachol or histamine. Steady-state responses of all concentrations were measured in each cell. Imaging and general data processing were performed as described in the previous section (see "FRET measurements" section). The FRET ratio $\Delta(F_{YFP}/F_{CFP})$ was normalized to the response at the maximum concentration set to 1. Concentration-response curves were fitted by GraphPad Prism 8.4 with variable slopes, the bottom constrained to 0 and the top to 1. For a detailed quantification, EC50 values were determined for each measurement and plotted relative to the native $G\alpha$ subunit in horizontal bars as ΔlogEC50. Absolute amplitudes at the maximum concentration were compared for every $G\alpha$ subunit against empty vectors (pcDNA3) instead of a transfected $G\alpha$ subunit to check for specific activation over endogenously expressed $G\alpha$ subunits.

**Statistics and reproducibility**. Results are represented as means ± SEM from at least three independent experiments (transfections) of n individual cells with $n \geq 7$ for all experimental groups. Data were only excluded if technical problems with the perfusion system occurred. Statistical analyses were performed using Graph-Pad Prism 8.4. One-tailed t-tests were used for comparison of two groups whereas one-way ANOVAs were used to compare more than two groups followed by Tukey's posttests to test each condition against the mean of every other condition or followed by Dunnett's posttest to compare each group against a control. Differences were considered statistically significant for $P < 0.05$. If differences were only one star significant ($0.01 \leq *P < 0.05$), measurements were requested to have $n \geq 10$ with a maximum difference of four n's between the lowest and highest sample size.

**Further methods**. Immunoblotting for the analysis of $G\alpha$ expression levels, the direct G protein activation assay, and the pertussis toxin pretreatment which were used for experiments shown in Supplementary Figures are described in Supplementary Methods.

**Reporting summary**. Further information on research design is available in the Nature Research Reporting Summary linked to this article.

## Data availability
The datasets generated and analyzed during this study which are not included in the Supplementary Data File are available from the corresponding author on reasonable request.

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

## Author contributions

V.J. and M.B. wrote the manuscript and designed the study and experiments. V.J. and N.M. designed and performed the experiments, M.B. supervised the study.

## Funding

## Competing interests

The authors declare no competing interests.
