## [Peer Review File · Communications Biology]

Reviewers' comments:

Reviewer #1 (Remarks to the Author):

Jelinek and Bünemann present an extensive investigation of G protein coupling specificity determinants using a chimeric approach and biophysical (FRET) techniques. The main conclusion of the manuscript is that selectivity determinants differ in importance for different receptors, but in general correspond to regions of the G α subunit that previous structural studies have identified as likely to be important. Notably, they specifically show that the distal C terminus, long thought to be a primary driver of specificity, is not as important as previously thought, at least for the muscarinic M3 acetylcholine receptor. The study is well-conceived and carried out with rigor (this was a lot of work), and the results are convincing. The number of receptors studied is relatively small, but the number of G protein chimeras used is quite large, and the data will be of interest to a wide audience. A few issues could, if addressed appropriately, significantly strengthen some of the conclusions and the presentation.

Specific comments:

One technical shortcoming of the manuscript is the lack of any experiments showing expression levels of the various G protein chimeras, which include many regions that may adversely affect protein stability. This is a less serious issue for the majority of chimeras that clearly support function, but is quite serious indeed for GqoC22, which is never shown to function or express in any way. Direct measurement of chimera expression by immunoblot would be ideal, but in lieu of this an indirect indication of expression and association with G betagamma subunits (which could be done with FRET) would be adequate. If chimera expression is not assessed, all results with Gqo22 are uninterpretable and should be removed, and the authors should state that expression of chimeras was not measured and was assumed to be roughly equal or not a factor.

The authors use an integration method (AUC) with an arbitrary time cutoff to compare receptor-G protein complex stability after agonist withdrawal, similar to that used in a previous publication by the same group. Some index is necessary and this is likely as good as any other, but the authors should not refer to this measurement as a "lifetime" (e.g. Fig. 1) or "binding affinity" (e.g. line 137) as it is neither. A precise nomenclature would be cumbersome, but the authors should use something more generic and all-encompassing (e.g. "stability").

The analysis relies on overexpression of G alpha subunits to the point where signals mediated by endogenous G proteins are not relevant. This is justified by showing that endogenous G alpha subunits are inadequate to pair with overexpressed G betagamma, but this should be mentioned earlier and more explicitly than Supplementary Figure S1b. On the same topic, the language on line 143 referring to "requesting" high amplitude responses is confusing. It should be replaced by a simple statement that endogenous G proteins did not support significant responses.

The authors allude to a sequence of events where "binding" precedes "activation". The distinction is important but overlooks the caveat that the complexes they are studying are at the end of the activation process, i.e. after GDP release. Recent work from the Kobilka group has suggested that nucleotide release may occur long before formation of a long-lasting complex of receptor and nucleotide-free G protein. In other words, the early association events (prior to nucleotide release) that determine specificity may not be exactly the same as those studied here. This possibility should be addressed in some manner in the discussion.

The final half-sentence of the manuscript (line 471) referring to structure-based drug design is at best unnecessary and should be removed. The results are sufficiently important without this implication.

Line 89- why is it surprising that Goq11 is similar to Go? Although this is contrary to some

expectations, essentially the same result has been published previously (ref. 29).

Line 216- "N-terminal alpha5" should be "N-terminal half of alpha5".

Reviewer #2 (Remarks to the Author):

In the manuscript, Jelinek et al. uses a sophisticated FRET- and microscope-based method to elucidate coupling selectivity determinants of G proteins for two Gq-coupled receptors (M3R and H1R). The authors conclude that various regions, especially outside of the C-terminal tail of the Ga subunit, collectively affect G-protein-coupling selectivity. Two complemented measurements (spontaneous dissociation kinetics of the GPCR-G protein complex in permeabilized cells and ligand-induced G protein activation in intact cells) provide a nice evidence for a correlation between the kinetics and the activation. The work is comprehensive and well described, and is suited for publication after a minor revision.

Major points:

In general, expression in the words sounds roundabout. The authors explain why they did and how they thought, etc., but it will become readable by describing more straightforward by trimming down such circumstantial sentences.

Line 49-52, the reference 6 describes coupling determinants outside of the Ga C-terminus. How do the authors position the reference? Does it lack experimental validation of "determinants"?

Figure 1, how do the authors ensure that in the authors' washing condition, G proteins are in a nucleotide-free state? Presence of GDP bound in G proteins would affect interpretations of the kinetics of the complex (e.g., Go has a fast GTPase and nucleotide exchange rate that Gq). The authors should use apyrase and confirm that the results are equivalent between the washing condition and apyrase treatment (also see the reference 29).

How does expression level of Ga constructs affect the lifetime and $\Delta\log EC_{50}$ values? For example, cellular responses A and B are not always linear. I assume that the $\Delta BRET$ value reflects an expression level, however, the authors should present equal expression levels of the constructs or a titration of the WT constructs to validate that the effect is not due to mere change in expression of the constructs.

Line 100-101, the authors should test the GqoC22-11 construct to demonstrate the "critical role of the N-terminal half of the $\alpha 5$ helix for binding to M3 receptors".

Minor points:

Line 17, I suggest including "of Ga" following "the $\alpha 5$ helix" to make the description clear to readers ("...the $\alpha 5$ helix of Ga to be key...").

Line 54, I suggest emphasizing the continuous work from the reference 4. For example, "Based on our previous finding that Gq proteins bind in a more stable manner than Go proteins to muscarinic M3 receptors (M3R) in the permeabilized cell experiment (ref4), in this study we aim at identifying..."

Line 141-145, G-protein-depleted HEK293 cells (3GKO plus PTX or 4GKO) may be useful for measuring a small chimeric G protein signal, which the authors describes as problematic.

<https://www.ncbi.nlm.nih.gov/pubmed/29362459>

<https://www.ncbi.nlm.nih.gov/pubmed/31147448>

Line 184, in my view at the M1R-G11 complex structure, the $\alpha 4$ - $\beta 6$ loop (the region uses not the Gq4 and the Gq4 mutants) looks a bit far from the receptor. How do the authors define interface in the Gq?

Line 199, is the flattening effect attributable to allosteric network? In my knowledge, there is a nice MD paper, which may explain the phenomenon.

<https://www.ncbi.nlm.nih.gov/pubmed/30289386>

Line 480-486, except for N- or C-terminal fused constructs, I recommend describing insertion site. This include YFP-fused Ga constructs.

Reviewer #3 (Remarks to the Author):

This paper from Jelinek and Bünemann identifies the regions of the Go and Gq alpha subunits which are important for the stability of the interaction between the G protein and GPCR, and the regions of the G alpha subunit which are important for controlling the strength of G protein activation by the receptor. They do this using two GPCRs that can couple to both Go and Gq, the muscarinic M3R and the histamine H1R. The stability of GPCR/G protein interaction is determined in single, live, permeabilised and nucleotide-depleted HEK293 cells by measuring FRET between a Citrine-tagged GPCR, and Turquoise-tagged G gamma subunit. The relative degree of activation of the G alpha subunit is determined in single, live HEK293 cells by measuring FRET between a Citrine-tagged GRK2 and the Turquoise-tagged G gamma subunit. Chimeric G proteins, where sections of Go have been replaced by the equivalent regions of Gq (and vice versa) were designed and generated. They find that, in general, the relative activation of a particular G alpha subunit is dictated by the stability of the receptor/G protein complex. The C-terminal region appears particularly important, with contributions from the N-terminus and the beta2/beta3 loop. The contributions of regions to stability and activation vary dependent upon the GPCR in question, and whether the G protein is a primary vs secondary signalling mediator.

These FRET experiments are very elegant and represent an enormous amount of work. The data are thoughtfully and very clearly presented, and the authors have included very clear and valuable depictions of the regions of the G alpha subunits that are switched in the chimeras.

Major comment:

1. Use of "binding affinity" to describe data from GPCR/G-gamma FRET dissociation experiments. These experiments show dissociation kinetics very nicely, but the calculated area under the curve is not an affinity value. The affinity of the G alpha for the receptor is not determined. Please change the terminology used throughout the manuscript – perhaps consider replacing binding "affinity" with "GPCR/G protein complex stability"?

Minor comments:

2. Supplementary Fig 4 shows very nice data to confirm that the "indirect" FRET assay reflects a more "direct" activation assay, and there is good rationalisation for use of the "indirect" method (i.e. no additional interference in the G alpha subunit by addition of a fluorescent tag). It would be useful to have the description of why the second stimulus & wash-out was used to monitor dissociation of the G protein from the receptor ("indirect" binding assay) when this set-up is first described in the results (as per methods, pg 27 lines 530-531).

3. State clearly somewhere early in the results precisely what you are using each FRET assay to

determine i.e. FRET between receptor and G-gamma is being interpreted as the binding of the G alpha subunit, FRET between GRK and G-gamma is being interpreted as the activation of the G alpha subunit.

4. Is there a significant difference between Goq2+C22 and GoqN+C22 in Fig 3C/D? By eye they look similar, but the text on pg 10 lines 195-196 implies that the M3-GoqN+C22 complex is more stable. Could an alternative interpretation be that Goq2+C11 reaches the resolution change limit of the chimeras (~100% change in AUC) such that there is no further slowing of dissociation with Goq2+C22? In contrast the GoqN+C11 does not reach this limit, and therefore the inclusion of the remaining C-tail residues in GoqN+C22 slows this further?

5. Observed curve flattening for GoqN+C11/GoqN+C22 with the M3R (Fig 3E) did not occur for these same chimeras at the H1R (Fig 5G). The authors showed clearly that the curve flattening was not due to endogenous Go proteins (Supp Fig 6). Given that this occurs for M3R and not H1R, could there be a contribution of a higher expression of endogenous M3R vs the H1R (4.70 vs 0.02 FPKM, respectively; Uhlen et al 2015 Science 347:1260419) in HEK293 cells, to this? The authors should add a brief discussion as to the cause of this as it is mentioned in the results.

6. Comparison of GqoC11 chimeras between M3R and H1R (pg 15, lines 305-308) – the authors show that GqoC11 dissociates much faster from H1R compared to M3R. This will, however, be inherently linked to the faster dissociation of wild-type Gq at the H1R compared to the M3R. Given that GqoC11 also dissociates faster from the M3R compared to wild-type Gq, I'm not sure how much should be made of this comparison. It should be easier to disrupt a less stable binding interaction (represented by the faster dissociation rate between wild-type Gq and H1R compared to M3R) – and this is exactly what is shown by comparing the data in Fig 5D to Fig 1D. As such, while it is clear that the 11 C-terminal amino acids of Gq contribute to stability of binding for both M3R and H1R, I'm not sure this data allows us to conclude that these residues are more important for one receptor over the other, as the stability of the receptor-Gq complexes are not the same to begin with.

7. Further to this – summary statement at end of results (pg 17 lines 348-349) "GqoC11 already impairs binding to H1R, whereas for M3R only the subsequent activation is compromised" This does not hold with Fig 1D which shows a significant acceleration in dissociation for GqoC11 at the M3R compared to wild-type Gq.

Revision: COMMSBIO-20-2151-T

Dear reviewers, we like to thank you very much for your extremely helpful comments. For the revision, which unfortunately took longer as expected, we tried our best to address all points raised by you and revised the manuscript accordingly. Please find below a detailed response to the individual comments by all reviewers. We set out to clarify all of your concerns and hope that the revised manuscript could be considered for publication.

Reviewer #1

- One technical shortcoming of the manuscript is the lack of any experiments showing expression levels of the various G protein chimeras, which include many regions that may adversely affect protein stability. This is a less serious issue for the majority of chimeras that clearly support function, but is quite serious indeed for GqoC22, which is never shown to function or express in any way. Direct measurement of chimera expression by immunoblot would be ideal, but in lieu of this an indirect indication of expression and association with G betagamma subunits (which could be done with FRET) would be adequate. If chimera expression is not assessed, all results with Gqo22 are uninterpretable and should be removed, and the authors should state that expression of chimeras was not measured and was assumed to be roughly equal or not a factor.
- We thank the reviewer for bringing up this important point. In the case where we observe functional responses with G protein chimeras, we don't consider expression levels of G α subunits as critical, since we rely not on amplitudes but rather on the kinetics of the signal. Specifically for the M₃R – G-protein complex dissociation after agonist withdrawal, the relative expression levels should not matter. Therefore, we stated in the results (line 83-85): "As the evaluation of kinetics rather than amplitudes should be independent of the relative G α expression levels, the expression was not determined as long as clear FRET-signals could be detected."

However, the reviewer is absolutely right for asking to demonstrate expression of GqoC22 as it didn't show significant responses in our assays. Therefore, we performed immunoblots of cells transfected with the exact same protocol as used for our FRET studies. A representative blot comparing the expression levels of GqoC22 and GqoC11 with Gq is now shown in Supplemental Fig. 1c. Furthermore, the proper expression of GqoC22-11, an additional chimera that was suggested by reviewer 2 which didn't interact with the receptor either is shown in Supplemental Fig. 1d.

- The authors use an integration method (AUC) with an arbitrary time cutoff to compare receptor-G protein complex stability after agonist withdrawal, similar to that used in a previous publication by the same group. Some index is necessary and this is likely as good as any other, but the authors should not refer to this measurement as a “lifetime” (e.g. Fig. 1) or “binding affinity” (e.g. line 137) as it is neither. A precise nomenclature would be cumbersome, but the authors should use something more generic and all-encompassing (e.g. “stability”).
- We totally agree with the reviewer about the inaccurate indication and are thankful for the suggestion to use a nomenclature like “stability”. Therefore, we replaced the words “lifetime” and “affinity” in the whole manuscript by “complex stability” or “binding stability”.
- The analysis relies on overexpression of G alpha subunits to the point where signals mediated by endogenous G proteins are not relevant. This is justified by showing that endogenous G alpha subunits are inadequate to pair with overexpressed G betagamma, but this should be mentioned earlier and more explicitly than Supplementary Figure S1b.
- We are glad to follow the reviewer’s suggestion by showing earlier in the manuscript that endogenously expressed G α subunits don’t lead to significant FRET signals which would disturb the measurement of overexpressed G proteins. Therefore, a pcDNA3 trace is now plotted in Fig. 1b showing that endogenous G α subunits lead to much lower amplitudes compared to overexpressed ones. We also mentioned in the results (line 77-78): “... whereas endogenous G α subunits (empty pcDNA3 vectors transfected instead of G α subunits) did not lead to noticeable FRET signals”.

- On the same topic, the language on line 143 referring to “requesting” high amplitude responses is confusing. It should be replaced by a simple statement that endogenous G proteins did not support significant responses.
- We agree with the reviewer that the significantly weaker activation of endogenous G proteins compared to overexpressed ones can be put more simply. Therefore, we stated in line 149-152: “Even though endogenous G proteins are also activated in this assay, which could not be prevented by a pertussis toxin (PTX) pretreatment (Supplementary Fig. 3c), chimeric G α subunits showed significantly higher FRET signals at the maximum carbachol concentration than cells transfected with empty vectors (pcDNA3) instead of G α subunits.”

- The authors allude to a sequence of events where “binding” precedes “activation”. The distinction is important but overlooks the caveat that the complexes they are studying are at the end of the activation process, i.e. after GDP release. Recent work from the Kobilka group has suggested that nucleotide release may occur long before formation of a long-lasting complex of receptor and nucleotide-free G protein. In other words, the early association events (prior to nucleotide release) that determine specificity may not be exactly the same as those studied here. This possibility should be addressed in some manner in the discussion.
- We thank the reviewer for pointing out that measurements of the receptor-G protein complex stability occur only after GDP release. Therefore, initial interactions (prior to nucleotide release) which might also contribute to coupling specificity cannot be assessed. For this reason, we stated in the discussion (line 425-427): "Early association events which might influence coupling specificity already before GDP release could, however, not be assessed." Nevertheless, our method enables the functional investigation of the coupling process at quite an early timepoint. Additionally, we specified in the discussion that the activation measurement follows the investigation of the G protein binding in a subsequent step which is reflected by the release of the activated $G\alpha$ subunit from the $G\beta\gamma$ subunit. Therefore, we stated in line 427-429: "In a subsequent step, we investigated how these complex stabilities were ultimately translated into physiologically relevant G protein activation reflected by the release of activated $G\alpha$ subunits from $G\beta\gamma$ subunits whose interaction with GRK2 was then analyzed."
- The final half-sentence of the manuscript (line 471) referring to structure-based drug design is at best unnecessary and should be removed. The results are sufficiently important without this implication.
- We agree with the reviewer that the final half-sentence does not contribute to the story of the manuscript and therefore we removed it.
- Line 89- why is it surprising that Goq11 is similar to Go? Although this is contrary to some expectations, essentially the same result has been published previously (ref. 29).
- We are glad about the note of the reviewer that a similar result was already revealed by a comparable study (ref. 29). The reason why we mentioned our finding as surprising is explained by the fact that ref. 29 was not yet published when we found out that GoqC11 behaved similar to Go. However, as ref 29 is published by now, we removed the word "surprisingly" at the beginning of the sentence in line 95: "After the withdrawal...".
- Line 216- "N-terminal alpha5" should be "N-terminal half of alpha5".
- We thank the reviewer for suggesting the specific phrase "N-terminal half of alpha5" instead of "N-terminal alpha5" which makes the text easier to understand. Therefore, we implemented this throughout the manuscript.

Reviewer #2

Major points:

- In general, expression in the words sounds roundabout. The authors explain why they did and how they thought, etc., but it will become readable by describing more straightforward by trimming down such circumstantial sentences.
- We thank the reviewer for the comment as it is in our own interest to make the text more readable. Therefore, long and nested sentences which occurred especially at the beginning of each section were now put more simply. Furthermore, we improved the language by specific suggestions (of the other reviewers as well) that should make the text more easily understandable.
- Line 49-52, the reference 6 describes coupling determinants outside of the G α C-terminus. How do the authors position the reference? Does it lack experimental validation of "determinants"?
- We are glad to follow the reviewer's suggestion to better position the interesting findings from reference 6 where coupling determinants over the whole G α subunit are predicted by an evolutionary approach. However, since there is a lack of conclusive biochemical studies especially for G α structures outside of the G α C-terminus, the hypothesized selectivity determinants lack experimental validation by functional studies. Therefore, we extended the sentence in line 52-54: "For this reason, there is a compelling need to unveil molecular details of selectivity determinants beyond the G α C-terminus which were so far only theoretically predicted for example by an evolutionary study⁶, but still lack experimental validation."
- Figure 1, how do the authors ensure that in the authors' washing condition, G proteins are in a nucleotide-free state? Presence of GDP bound in G proteins would affect interpretations of the kinetics of the complex (e.g., G α has a fast GTPase and nucleotide exchange rate that G $\beta\gamma$). The authors should use apyrase and confirm that the results are equivalent between the washing condition and apyrase treatment (also see the reference 29).
- We agree with the reviewer's argumentation that nucleotide-free G proteins are critical for the measurement of the receptor-G protein complex stability. In a previous publication (reference 4, Fig. 1e,f; screenshot below) in which we described the assay used in the present study, we very carefully addressed this question and indeed we observe the nucleotide-free complex. Some arguments are shown below: There were remarkable differences in amplitudes and kinetics between intact (E, left) and permeabilized nucleotide-depleted cells (F, right) as without nucleotides G $\beta\gamma$ proteins dissociated much slower from the M $_3$ R (within minutes instead of seconds) which is consistent with the present study. Moreover, as the presence of GDP bound in G proteins will only generate small FRET signals, they should not contribute a lot to the overall stronger signal of nucleotide-free receptor G protein complexes.

- Moreover, in reference 4 (Supplemental Fig. S3a,b; screenshot below) adequate nucleotide depletion was also proved by the addition of low doses of 100 nM GTP (A, left) and 100 nM GDP (B, right) which led to the distinctly faster dissociation of Gαq from the M₃R compared to buffer (if no other substance is indicated above the graph). These kinetics prove indeed the previous nucleotide-free state. 100 μM GTP or GDP further accelerated the off-rate of Gαq from the M₃R and mimicked the conditions of Gq binding to M₃R in intact cells (reference 4, Fig. 1e; screenshot above). Furthermore, the laminar perfusion with buffer was able to wash out the nucleotides as the stimulation with 10 μM ACh in the end led to similar FRET-signals and off-rates as in the beginning of the experiment. Therefore, nucleotide-free conditions were experimentally validated.

To refer to the previous publication, we added the following sentence to the methods section (line 546-548): “The permeabilization process was optimized by an adequate saponin concentration and incubation time followed by sufficient washing steps to ensure nucleotide-free conditions as previously demonstrated⁴”. Moreover, it is now mentioned in line 81-83 that the examination of the second washout phase ensured the continuous depletion of nucleotides by the perfusion system. All in all, we suggest that an apyrase pretreatment would lead to comparable results.

- How does expression level of G α constructs affect the lifetime and $\Delta\log EC_{50}$ values? For example, cellular responses A and B are not always linear. I assume that the $\Delta BRET$ value reflects an expression level, however, the authors should present equal expression levels of the constructs or a titration of the WT constructs to validate that the effect is not due to mere change in expression of the constructs.
- We agree with the reviewer that expression levels and amplitudes ($\Delta FRET$) are somehow connected. However, as mentioned in the answer to reviewer #1, we don't consider expression levels of G α subunits as critical, since we rely not on amplitudes but rather on the kinetics of the signal. Specifically for the M $_3$ R – G-protein complex dissociation after agonist withdrawal, the relative expression levels should not matter. Therefore, we stated in the results (line 83-85): “As the evaluation of kinetics rather than amplitudes should be independent of the relative G α expression, the expression levels were not determined as long as clear FRET-signals could be detected.” Furthermore, immunoblots requested by reviewer #1 are now shown in Supplemental Fig. 1c,d and support similar expression levels at least for Gq α C chimeras compared to Gq.

Moreover, the calculated $\Delta\log EC_{50}$ values depend on the FRET-signal generated by the G $\beta\gamma$ - GRK2 interaction. As G α constructs necessary for functional heterotrimers were transfected in excess (1.5 μ g G α , 0.5 μ g G β , 0.2 μ g G γ) we suggest that a variable G α expression would not affect the results. Besides, data were obtained from a group of cells (about 5 to 10) and thus, the expression levels of GRK2 and G $\beta\gamma$ which limit the FRET signal should be averaged out.

If G α chimeras are potentially very weakly expressed, G $\beta\gamma$ might also pair with endogenous G α subunits. As a control, all functional G α chimeras led to significantly higher FRET signals than without transfected G α subunit (pcDNA3) (Fig. 2a,b; Supplemental Fig. 7), there should be barely any endogenous interference. Only the GqC22 construct activated by the H $_1$ R appeared slightly below the significance limit which was, however, discussed in line 336-338: “However, the small amplitude may explain the slightly lower potency gain compared to the M $_3$ R due to competition with less potent endogenous G proteins.” Taken together, our $\Delta\log EC_{50}$ values should not be critically affected by a variable G α expression.

- Line 100-101, the authors should test the GqoC22-11 construct to demonstrate the "critical role of the N-terminal half of the $\alpha 5$ helix for binding to M₃ receptors".
- We thank the reviewer for the good suggestion to investigate the GqoC22-11 construct which will undoubtedly complete the story. Therefore, we analyzed this newly cloned construct which, however, did not interact with the M₃R similar to GqoC22 since the amplitude of the new chimera was comparable to the level of pcDNA3 (empty vector instead of G α subunit). This is now illustrated in Supplementary Fig. 1b. Additionally, we stated in line 105-108: "... the exchange of the N-terminal half of the $\alpha 5$ helix in GqoC22-11 and the full exchange in GqoC22 completely prevented binding (Supplementary Fig. 1b) even though the proper expression could be confirmed by immunoblots (Supplementary Fig. 1c,d). Therefore, the N-terminal half of the $\alpha 5$ helix seems to play a critical role for binding to M₃ receptors."

Minor points:

- Line 17, I suggest including "of G α " following "the $\alpha 5$ helix" to make the description clear to readers ("...the $\alpha 5$ helix of G α to be key...").
- We thank the reviewer for this useful suggestion which makes the sentence more easily comprehensible. Therefore, we implemented "of G α " in the abstract accordingly.
- Line 54, I suggest emphasizing the continuous work from the reference 4. For example, "Based on our previous finding that Gq proteins bind in a more stable manner than Go proteins to muscarinic M₃ receptors (M₃R) in the permeabilized cell experiment (ref4), in this study we aim at identifying..."
- We are glad to take over the proposed sentence to highlight the continuous work based on our previously developed assay. So we stated in line 55-56: "Based on our previous finding that in permeabilized cells Gq proteins bind in a more stable manner than Gi/o proteins to muscarinic M₃ receptors (M₃R)⁴, this study aimed at identifying..."
- Line 141-145, G-protein-depleted HEK293 cells (3GKO plus PTX or 4GKO) may be useful for measuring a small chimeric G protein signal, which the authors describe as problematic.
<https://www.ncbi.nlm.nih.gov/pubmed/29362459>
<https://www.ncbi.nlm.nih.gov/pubmed/31147448>
- We agree with the reviewer's suggestion that cells lacking certain endogenous G protein classes might help for measurements of some G α chimeras whose weak signals are difficult to detect next to endogenously expressed G α subunits. However, since most of the chimeras behaved fine in our assays and it is not feasible in a short amount of time to repeat experiments in these KO-cells, the new approach might be rather useful for future experiments.

- Line 184, in my view at the M1R-G11 complex structure, the α 4- β 6 loop (the region uses rot the Goq4 and the Gqo4 mutants) looks a bit far from the receptor. How do the authors define interface in the Gq?
 - The reviewer is right about the α 4- β 6 loop (in Goq4 and Gqo4 chimeras) which looks a bit far away from the M₁R in the M₁R-G11 complex structure. However, the ICL3 located between TM5 and TM6 is not resolved in this structure although containing about 200 amino acids for the M₃R. Therefore, it might still interact with the α 4- β 6 loop. We now stated in line 197-198: “Unfortunately, the ICL3 is not resolved in the structure but might still interact with the α 4/ β 6-loop as the ICL3 is quite big for muscarinic receptors.” Furthermore, reference 30 mentions several studies that implicate this loop to be in contact with the receptor.

- Line 199, is the flattening effect attributable to allosteric network? In my knowledge, there is a nice MD paper, which may explain the phenomenon.
 - We thank the reviewer for proposing an idea which might explain the flattening effect of some concentration response curves in the G $\beta\gamma$ - GRK2 interaction assay. We are aware that the allosteric networks of Gq and GoqN+C22 might differ from each other as they are based on distinct G protein backbones. However, we cannot imagine how the allosteric network might influence the G protein activation potency in a way that low agonist concentrations will lead to a relatively strong activation compared to a relatively weak activation for higher concentrations (resulting in flatter concentration-response curves). Therefore, we assume that there must be another reason for the flattening effect.

- Line 480-486, except for N- or C-terminal fused constructs, I recommend describing insertion site. This include YFP-fused G α constructs.
 - We thank the reviewer for the recommendation to include insertion sites of fluorophores if they are not N- or C-terminal fused to the protein. We now exactly describe where the fluorophores are inserted in the respective G α -YFP constructs in line 501-502: “...G α q-YFP (YFP inserted between F124 and E125)³³ ...and G α o-YFP (YFP inserted between L91 and G92)³⁴ and...”.

Reviewer #3

Major comment:

- 1. Use of “binding affinity” to describe data from GPCR/G-gamma FRET dissociation experiments. These experiments show dissociation kinetics very nicely, but the calculated area under the curve is not an affinity value. The affinity of the G alpha for the receptor is not determined. Please change the terminology used throughout the manuscript – perhaps consider replacing binding “affinity” with “GPCR/G protein complex stability”?
- We completely agree with the reviewer’s argumentation. Therefore, we replaced “binding affinity” with “complex stability” or “binding stability” throughout the manuscript.

Minor comments:

- 2. Supplementary Fig 4 shows very nice data to confirm that the “indirect” FRET assay reflects a more “direct” activation assay, and there is good rationalisation for use of the “indirect” method (i.e. no additional interference in the G alpha subunit by addition of a fluorescent tag). It would be useful to have the description of why the second stimulus & wash-out was used to monitor dissociation of the G protein from the receptor (“indirect” binding assay) when this set-up is first described in the results (as per methods, pg 27 lines 530-531).
- We thank the reviewer for suggesting that the rationale of evaluating the second wash-out phase should already be explained when the assay is firstly introduced in the results. Therefore, we now stated in line 81-83: “Furthermore, the examination of the second decay ensured the entire depletion of potential remaining nucleotides as they were continuously purged away by the perfusion.”
- 3. State clearly somewhere early in the results precisely what you are using each FRET assay to determine i.e. FRET between receptor and G-gamma is being interpreted as the binding of the G alpha subunit, FRET between GRK and G-gamma is being interpreted as the activation of the G alpha subunit.
- We are glad to accept the reviewer’s suggestion to clearly explain how we interpret our measurements as this should help to understand the story. Therefore, we modified the introduction of the respective assay as follows:
Line 72-74: „Interactions between YFP-labeled receptors and CFP-labeled G γ_2 subunits interpreted as G protein binding to the receptor were measured upon agonist stimulation in single permeabilized HEK293T cells by means of FRET as illustrated in Fig. 1a.”
Line: 138-141: “For this reason, FRET-based measurements were performed by analyzing the recruitment of YFP-labeled GRK2 by CFP-labeled $\beta\gamma$ subunits which had to be dissociated from activated G α subunits. Thus, the G α activation could be investigated indirectly while stimulating the M $_3$ R with increasing concentrations of carbachol (representative cells; Supplementary Fig. 3a,b).”

- 4. Is there a significant difference between Goq2+C22 and GoqN+C22 in Fig 3C/D? By eye they look similar, but the text on pg 10 lines 195-196 implies that the M₃-GoqN+C22 complex is more stable. Could an alternative interpretation be that Goq2+C11 reaches the resolution change limit of the chimeras (~100% change in AUC) such that there is no further slowing of dissociation with Goq2+C22? In contrast the GoqN+C11 does not reach this limit, and therefore the inclusion of the remaining C-tail residues in GoqN+C22 slows this further?
 - We thank the reviewer for the new idea to interpret the results. As the AUC of GoqN+C22 in Fig. 3c was on average increased by 118% whereas the AUC of Goq2+C22 in Fig. 3d increased only by 99%, both G α chimeras might still be different. However, we haven't statistically compared both conditions against each other and therefore, we removed the following phrase in line 204: "...and thus did not keep up with the stability of M₃R-GoqN+C22 complexes". Furthermore, we agree with the reviewer that there might be a resolution change limit which is clearly not reached for GoqN+C11 so that GoqN+C22 can be observed to dissociate even slower. In contrast, Goq2+C11 is already close to a ~100% change in AUC so that Goq2+C22 might have less potential to additionally slow down the dissociation. However, we assume that identical Go backbones should exhibit similar resolution limits which means that Goq2+C22 might still have the opportunity to increase the AUC change from 100% to 120%. Moreover, at the H₁R Goq2+C22 showed even a less stable binding than Goq2+C11 further suggesting that the construct is probably not restrained by a resolution limit with the M₃R.
- 5. Observed curve flattening for GoqN+C11/GoqN+C22 with the M₃R (Fig 3E) did not occur for these same chimeras at the H₁R (Fig 5G). The authors showed clearly that the curve flattening was not due to endogenous Go proteins (Supp Fig 6). Given that this occurs for M₃R and not H₁R, could there be a contribution of a higher expression of endogenous M₃R vs the H₁R (4.70 vs 0.02 FPKM, respectively; Uhlen et al 2015 Science 347:1260419) in HEK293 cells, to this? The authors should add a brief discussion as to the cause of this as it is mentioned in the results.
 - We agree with the reviewer about the fact that curve flattening for GoqN+C11 / GoqN+C22 is only observed with the M₃R and not with the H₁R. However, even if the endogenous expression of M₃R would be higher than for H₁R, we suggest that this should not affect the overall FRET signal as in our assay both receptors are much more overexpressed by the transfection. Moreover, we cannot imagine how a slightly higher M₃R expression might selectively flatten the slope of some Go-based chimeras. If we have to further speculate about the reason, there are two phenomena which might be connected to these observations. Firstly, activated Gq subunits (and possibly also GoqN+C11 / GoqN+C22 chimeras) can recruit GRK2 by themselves even at low agonist concentrations (Wolters et al., 2014, Molecular Pharmacology), whereas Gi/o subunits are probably not able to promote GRK2 recruitment. Furthermore, Gq proteins can preassociate with a polybasic cluster in the M₃R C-terminus (Qin et al., 2011, Nature Chemical Biology) which does not exist in the C-terminus of the H₁R. These theories might potentially explain a G protein and receptor specific difference even though we are not able to directly link them to the curve flattening effect. We now added a brief discussion about the issue of curve flattening (line 473-478).

- 6. Comparison of GqC11 chimeras between M3R and H1R (pg 15, lines 305-308) – the authors show that GqC11 dissociates much faster from H1R compared to M3R. This will, however, be inherently linked to the faster dissociation of wild-type Gq at the H1R compared to the M3R. Given that GqC11 also dissociates faster from the M3R compared to wild-type Gq, I'm not sure how much should be made of this comparison. It should be easier to disrupt a less stable binding interaction (represented by the faster dissociation rate between wild-type Gq and H1R compared to M3R) – and this is exactly what is shown by comparing the data in Fig 5D to Fig 1D. As such, while it is clear that the 11 C-terminal amino acids of Gq contribute to stability of binding for both M3R and H1R, I'm not sure this data allows us to conclude that these residues are more important for one receptor over the other, as the stability of the receptor-Gq complexes are not the same to begin with.
 - We agree with the reviewer's opinion that it might be easier to disrupt a slightly weaker H1R-complex by mutations than a more stable M3R complex. However, we show absolute AUC values for M3R-Gq (Supp. Fig. 2: 1d; **AUC \approx 110**) and H1R-Gq (Supp. Fig. 2: 5d; **AUC \approx 100**) demonstrating that Gq binds actually very stable at both receptors with not such a big difference. In contrast, the relative AUC decrease of H1R-GqC11 (Fig. 5d; 12%) was roughly 3-fold higher than for M3R-GqC11 (Fig. 1d; 4%) suggesting that the 11 C-terminal amino acids might be still more important for binding to the H1R. However, we cannot exclude the possibility that the slight tendency of more stable M3R-Gq complexes leads to lower effects of GqC11 at the M3R. Therefore, we now are a bit more cautious and state in line 315-317: "For Gq-based chimeras, there was a noticeable difference between both receptors, as GqC11 dissociated distinctly faster than Gq from the H₁R (Fig. 5d) whereas the difference is less pronounced for the M₃R (Fig. 1d)."
- 7. Further to this – summary statement at end of results (pg 17 lines 348-349) "GqC11 already impairs binding to H1R, whereas for M3R only the subsequent activation is compromised" This does not hold with Fig 1D which shows a significant acceleration in dissociation for GqC11 at the M3R compared to wild-type Gq.
 - We thank the reviewer for pointing out the confusing wording. We agree that we shouldn't ignore the significant faster dissociation for M3R-GqC11 complexes in Fig. 1d even though the AUC was only decreased by ~4%. Therefore, the sentence now takes into account that the binding of GqC11 at the M3R is at least slightly altered (line 360-361): "... GqC11 clearly impairs the binding stability at the H₁R, whereas for M₃R the subsequent activation of GqC11 is considerably stronger compromised than the only slightly altered binding."

REVIEWERS' COMMENTS:

Reviewer #1 (Remarks to the Author):

The authors have done an excellent job responding to all of my comments. They are to be congratulated on an excellent study.

Reviewer #2 (Remarks to the Author):

While some of my requests were not fully addressed by experimentation, the authors nicely addressed all of my concerns with new data, additional description and explanation of previously published data.

Reviewer #3 (Remarks to the Author):

The authors have responded to my comments in a very thoughtful and thorough way. In my view this paper is suitable for publication. I look forward to seeing it in print.